# Cellulase Enzyme Production from Filamentous Fungi *Trichoderma reesei* and *Aspergillus awamori* in Submerged Fermentation with Rice Straw

**DOI:** 10.3390/jof7100868

**Published:** 2021-10-16

**Authors:** Laila Naher, Siti Noor Fatin, Md Abdul Halim Sheikh, Lateef Adebola Azeez, Shaiquzzaman Siddiquee, Norhafizah Md Zain, Sarker Mohammad Rezaul Karim

**Affiliations:** 1Faculty of Agro-Based Industry, Universiti Malaysia Kelantan Jeli Campus, Jeli 17600, Malaysia; fatinrahim98@gmail.com (S.N.F.); abdulhalimsheikh@gmail.com (M.A.H.S.); lateef.aa@unilorin.edu.ng (L.A.A.); norhafizah.mz@umk.edu.my (N.M.Z.); 2Institute of Food Security and Sustainable Agriculture, Universiti Malaysia Kelantan Jeli Campus, Jeli 17600, Malaysia; 3Institute of Research and Poverty Management (InsPek), Universiti Malaysia Kelantan Bachok, Bachok 16400, Malaysia; 4Department of Plant Biology, Faculty of Life Sciences, University of Ilorin, Ilorin 240003, Nigeria; 5Biotechnology Research Institute, Universiti Malaysia Sabah, Jalan UMS, Kota Kinabalu 88400, Malaysia; shafiqpab@ums.edu.my; 6Faculty of Agriculture, Universiti Putra Malaysia, UPM, Serdang 43400, Malaysia; rkarimbau@yahoo.com or

**Keywords:** cellulase enzyme, *Aspergillus awamori*, *Trichoderma reesei*, rice straw, fermentation

## Abstract

Fungi are a diverse group of microorganisms that play many roles in human livelihoods. However, the isolation of potential fungal species is the key factor to their utilization in different sectors, including the enzyme industry. Hence, in this study, we used two different fungal repositories—soil and weed leaves—to isolate filamentous fungi and evaluate their potential to produce the cellulase enzyme. The fungal strains were isolated using dichloran rose bengal agar (DRBA) and potato dextrose agar (PDA). For cellulase enzyme production, a rice straw submerged fermentation process was used. The enzyme production was carried out at the different incubation times of 3, 5, and 7 days of culture in submerged conditions with rice straw. Fungal identification studies by morphological and molecular methods showed that the soil colonies matched with *Trichoderma reesei,* and the weed leaf colonies matched with *Aspergillus awamori*. These species were coded as *T. reesei* UMK04 and *A. awamori* UMK02, respectively. This is the first report of *A. awamori* UMK02 isolation in Malaysian agriculture. The results of cellulase production using the two fungi incorporated with rice straw submerged fermentation showed that *T. reesei* produced a higher amount of cellulase at Day 5 (27.04 U/mg of dry weight) as compared with *A. awamori* (15.19 U/mg of dry weight), and the concentration was significantly different (*p* < 0.05). Our results imply that *T. reesei* can be utilized for cellulase production using rice straw.

## 1. Introduction

Nature is blessed with an abundance of microorganisms, the majority of which are beneficial for our environment. Among these, the fungal kingdom plays a vital role in agriculture by decomposing by-products and managing environmental waste. The discovery of new fungal species or strains has drawn the attention of researchers to cellulase enzyme production, as it is one of the most important industrial enzymes.

Cellulase enzymes are used in various industries, such as laundry and detergent, textile production, pulp and paper, foods, and biofuel production. Oxidative and hydrolytic enzymes are used by fungal cellulases to degrade cellulose in plant biomass [1]. In the conversion of biomass and other waste material, the importance of cellulases has been widely studied over the past few decades. Cellulases are the most important enzymes used in the textile industry [2]. The application of these enzymes gives fabrics greater softness, as they cause fibers to protrude, and reduce the appearance of fading in fabrics [3]. Although commercial cellulase enzymes greatly improve performance via their use in industrial processes, the hydrolysis process used to manufacture the enzymes has a high operational cost—the highest cost in the enzyme industry [4]. Agricultural biomass has thus attracted attention for use as a substrate, or carbon source, to produce cellulase enzymes under submerged fermentation using microbial species [5].

Rice straw generated during the harvesting and milling process is an abundant agricultural biomass product in rice-growing countries but is not fully utilized as a by-product [6]. Rice straw has great potential as a biomass rich in cellulose (39%), hemicellulose (22%), lignin (16%), and ash (18%) [7]. It is a non-starch-based fibrous part of the plant. As mentioned earlier, cellulase enzymes are important in the enzyme industry [8]. Hence, rice straw can be used as a good source to produce cellulase enzymes. As the production of cellulase enzymes is costly, the use of rice straw could provide a cheaper substrate for a fermentation process incorporated with microbes to reduce the overall cost [9]. Cellulase enzymes are released from microbes, such as bacteria and fungi, by the mechanism of decomposition or by the fermentation process (Ahmed and Bibi, 2018) [9]. Hence, a good microbial agent is needed to enhance the production of cellulase enzymes. Therefore, in this study, we aimed to isolate local fungal species and use them for cellulase production in submerged fermentation with rice straw.

## 2. Materials and Methods

### 2.1. Sample Collection

For this study, we chose two different materials—soil samples and weed plants—for new fungal species or strain isolation. Paddy, banana, and rubber soil rhizosphere samples at Kelantan, Malaysia, were collected to isolate the potential bioactive fungal species of *Trichoderma*. About 100 g of soil was collected, at a depth of 10 cm from the surface, using a soil auger. After collection, the soil was immediately transferred into a plastic zipper bag and then placed into a cooler box until it reached the lab.

For endophytic fungal isolation, the leaves of the weed, *Parthenium hysterophorus,* were collected from the Durian Tunggal agricultural area, Melaka, Malaysia. After collection, the leaves were kept in a chiller box at 4 °C for storage. The samples of rice straw were collected from the Malaysian Agricultural Research and Development Institute (MARDI) paddy field, Seberang Perai, Pulau Pinang, Malaysia.

### 2.2. Medium Preparation

Three different media were used in this study: dichoran rose bengal chloramphenicol (DRBC) as a special medium for soil fungus *Trichoderma* spp. isolation, potato dextrose agar (PDA), and potato dextrose broth (PDB) for the weed endophytes. To prepare these media, 31.6 g of DRBC, 19.5 g of PDA, or 12 g of PDB powder were weighed and transferred into 500 mL of distilled water in a conical flask, separately for each medium. All conical flasks were heated to dissolve the mixture completely using a hot plate stirrer. Once the media were homogenized, they were sterilized by autoclaving at 121 °C, 15 psi for 15 min. After autoclaving was complete, and the bottle cooled down, 0.05 g of streptomycin sulphate was added into each medium’s bottle to provide bacterial resistance. 

### 2.3. Preparation of Soil Suspensions and Cultures

To prepare the soil suspensions, 10 g of each soil sample was weighed using an electrical balance (Sartorius, Malaysia). Then, the 10 g soil samples were diluted in 100 mL of sterile water in conical flasks and mixed using a hot plate stirrer with the aid of a magnetic stirrer at 100 rpm at room temperature for 10 min. Next, soil serial dilutions were prepared from the soil suspensions to isolate the *Trichoderma* colonies from the soils. One milliliter of soil suspension was transferred into a test tube containing 9 mL of autoclaved distilled water to make a 10^−1^ dilution. The serial dilution was repeated to make 10^−2^, 10^−3^, 10^−4^, and 10^−5^ dilutions. Under sterile conditions, 1 mL of each dilution (10^−1^, 10^−2^, 10^−3^, 10^−4^, and 10^−5^) was pipetted into a Petri dish, followed with about 9 mL of DRBC agar solution to make a soil culture [10]. The culture plates were made in three replicates, and all plates were incubated in a fungal incubator at 27 °C to isolate *Trichoderma* colonies.

### 2.4. Preparation of Parthenium Weed Leaves for Endophytic Fungi Isolation

The leaves of Parthenium *(Parthenium hysterophorus)* were first cleaned with a soft brush and then cut into small pieces of about 2 cm^2^ each. The leaf pieces were transferred into 10% Clorox for 30 s. This step was repeated three times; then, in the final repeat of the step, the leaf pieces were transferred onto tissue paper to dry the leaves. Next, they were transferred onto PDA medium plates, the cultures were made in three replicates, and all plates were incubated in a fungal incubator at 27 °C for three to five days.

### 2.5. Isolation of Fungi and Pure Cultures

The diverse fungal growth from the soil cultures on DRBA agar plates was observed daily and the colonies formed were calculated. The visible fungal colonies that formed were identified, based on their macromorphology characteristics, for the primary selection of *Trichoderma* colonies, and then isolated in a PDA medium to obtain a pure culture of fungus. The fungal colonies from Parthenium leaves were very imperfect and appeared as cloud colonies. However, only two colonies were chosen for pure culture in the PDA medium. 

### 2.6. Fungal Colony Identification by a Morphological Technique

Colony characteristics are usually not sufficiently precise for characterization as they do not provide enough information to establish species identification. Therefore, colony morphological features, including the shapes and sizes of conidia, the branching patterns of conidiophores, the shapes and sizes of phialides, and the production of chlamydospores, were used to carry out an anatomical study via slide culture observation.

#### Slide Culture Preparation

Slide cultures were prepared for the identification of the fungal colonies from the soil and leaf samples. For the preparation of the slide cultures, about 6 mm × 6 mm of PDA was cut and placed on the center of the microscopic slide. The four sides of the agar square were inoculated with fungal mycelia and covered with a microscope cover slip. The cover slips were placed in Petri dishes with a water-soaked cotton swab and filter paper to keep the environment in the Petri dish moist for fungal growth. The Petri dish was covered and incubated at room temperature for two days to allow the mycelia to grow. 

After two days, a thin section of the agar slide culture that contained the fungus was placed on a clean microscopic slide for identification based on its microscopic characteristics, such as conidia, conidiophores, phialides, pigmentation, and spore structure, observed under a light microscope (Leica, Kuala Lumpur, Malaysia) to identify the filamentous fungal species following the key of Aneens et al. [11]. 

### 2.7. Molecular Identification

#### Fungal DNA Extraction, Amplification, and Sequencing

Seven-day-old pure cultures of the morphologically similar isolates were used for genomic DNA extraction. DNA extraction was carried out using a commercial kit (Apical, Kuala Lumpur, Malaysia) following the manufacturer’s instructions. The polymerase chain reaction (PCR) was carried out using the primer pair ITS1 (5′-TCCGTAGGTGAACCTGCGG-3′) and ITS4 (5′-TCCTCCGCTTATTGATATGC-3′) [12] to amplify the internal transcribed spacer (ITS) region in the DNA using a thermocycler (Eppendorf, Hamburg, Germany). The PCR mixture was 25 µL and was prepared by adding 12.5 µL of Taq 2X PCR master mix, 1 µL each of primers ITS 1 and ITS 4 (10 µM), 9.5 µL of double-sterilized distilled water (ddH_2_O), and 1 µL of the DNA template with ddH_2_O, used as the template for the control reaction instead of the DNA. The PCR program used was 2 min at 94 °C, followed by 35 cycles at 94 °C for 1 min, 55 °C for 1 min, and 72 °C for 1 min, with final extension at 72 ° C for 10 min. Sanger sequencing of the PCR products was done at 1st BASE Asia Sdn Bhd (Selangor, Malaysia) using the same forward and reverse primers.

The results of the sequencing were used for similarity searches on the GenBank database using the BLAST-n tool (http://blast.ncbi.nlm.nih.gov, accessed on 29 August 2021). Similar sequences and outgroups were downloaded and aligned in AliView 1.17 [13]. The phylogenetic tree was generated in MEGA-X [14] using maximum parsimony analysis with 1000 bootstrap replications. 

### 2.8. Counting Colonies for Fungal Population Density

Seven-day pure cultures of *Trichoderma* sp. and the endophytic fungus of *Awamori* sp. were used for colony counting. For conidia counting, 10 mL of distilled water was added into the culture plates, scraped gently using an L-shaped rod, then poured into a slant bottle as a stock suspension for the preparation of serial dilutions for conidia counting. A volume of 1 mL of the stock suspension was transferred into a new slant bottle with 9 mL of distilled water and shaken gently; the process was repeated for dilutions of 10^−2^, 10^−3^, 10^−4^, and 10^−5^. The colony density was determined for each dilution by pipetting 1 mL of the mixture onto the cover slip of a hemocytometer on both sides. 

The conidia were counted using the equation from Louis et al. [15], as shown below:

Total viable cells = Grid 1 + Grid 2 + Grid 3 + Grid 4 + Grid 5

% of viable cells: (total viable cells)/(total of cells) × 100%

Average of total cells: (viable cells)/(squares (grid)) 

Dilution factor: (final volume)/(initial volume) 

Concentration of viable cells (conidia/μL):

=average total cells/squares × dilution factor × volume:

Grid: A framework of spaced bars in the hemocytometer;

Viable cell: A cell that appears in the hemocytometer grid;

Final volume: Total volume of distilled water used for the suspension;

Dilution factor: Ratio of the initial volume to the final volume.

### 2.9. Preparation of Rice Straw as a Pretreatment for Cellulose Extraction

Rice straw was used as a carbon source for the fungal isolate to produce cellulase enzyme. Before fermentation of the rice straw, it was pretreated with NaOH. In brief, rice straw was cut into small pieces for easy handling and soaked in NaOH (1N) for 12 h at room temperature. After two days of fermentation, the rice straw was sundried for 3–4 days, ground with a grinding machine, and then kept in a zipper bag for future use.

### 2.10. Cellulase Enzyme Production and Extraction

Cellulase production was done using the isolated fungi of *Trichoderma* sp. from soil, and *Aspergillus* sp. from parthenium leaf, in submerged formation with 30 g dry weight of rice straw for cellulose extraction in an Erlene Meyer flask. Before inoculation with the rice straw, a nutrient solution of potato dextrose broth (PDB) medium was prepared. The medium was prepared by weighing 12 g of PDB powder, mixing it together with 500 mL of distilled water in a conical flask, and autoclaving it for 1 h 30 min at 121 °C. Then, 10 mL suspension cultures of *Trichoderma* sp. (3.8 × 10^6^ conidia/μL), and *Aspergillus* sp. (1.908 × 10^6^ conidia/μL), were added into separate flasks containing 100 mL PDB. At this stage, enzyme production via fermentation with both fungi was conducted under varying incubation times of T1, 3 days; T2, 5 days; and T3, 7 days.

The 100 mL PDB cultures, containing 30 g of rice straw with a suspension of *Trichoderma* sp. or *Aspergillus* sp., were used for enzyme extraction. The broth culture was filtered using a filter funnel to separate the mycelia from the culture broth and then centrifuged at 6000× *g* for 10 min at 4 °C to remove any remaining mycelia. The collected supernatant was used for the enzyme assay.

### 2.11. Cellulase Enzyme Assay 

The enzyme assay was carried out on each *T. reesei* and *A. awamori* sample for the three incubation times at 3, 5, and 7 days post-incorporation (DPI) using the method of Maftukhah and Abdullah [16]. In brief, a rolled Whatman no.1 filter paper strip, of the dimensions 1.0 × 6 cm (50 mg), was placed into each assay tube. The filter paper strip was saturated with 0.1 M of sodium citrate buffer (pH 4.8) and equilibrated for 10 min at 50 °C in a water bath. Appropriately diluted enzymes (supernatants) in different amounts (500 μL, 400 μL, 300 μL, 250 μL, 200 μL, and 100 μL) were added to each tube to a final volume of 1 mL using distilled water and incubated for 60 min at 50 °C in a water bath. The experiment was carried out in three replicates. The reaction was stopped by adding 3 mL of 3,5-Dinitrosalicylic acid reagent per tube. Tubes were then incubated for 5 min in a boiling water bath for color development and cooled rapidly on ice. The reaction mixture was diluted appropriately and was measured against a reagent blank at 540 nm in a UV–vis spectrophotometer (Spectroquant pharo 300, Merck, Darmstadt, Germany).

### 2.12. Enzyme Assay Analysis

To analyze the cellulase enzyme activity, a standard curve was generated from standard glucose enzyme [17], with concentrations of 1, 2, 2.5, 3, 3.5, 4, 4.5, and 5.0 mg. Then, the activity was analyzed using the standard curve equation (where Y = sample OD reading, 0.0027 = rate of change of standard curve, x = total glucose activity, and 0.0178 = intercept of curve line), which was generated from the standard curve obtained from the simple linear regression model. Cellulase activity is expressed herein in units of cellulase activity per milligram of rice straw dry weight per hour [18]. 

## 3. Results

### 3.1. Fungal Colony Isolation from Soil and Weeds

The soil fungus isolation was carried out using a DRBC medium for the *Trichoderma* species. A total of 355 colonies were grown from paddy, banana, and rubber plantation soil suspension cultures. Although we used a serial dilution factor of 10 for the soil cultures, the colonies from different locations grew in different factors. Colonies appeared only on dilutions of 10^−1^ and 10^−2^ for paddy fields, 10^−3^ for the rubber plantation, and 10^−1^ for the banana plantation. The colonies on Petri dishes that showed yellowish and greenish mycelia on DRBC agar were selected as *Trichoderma* colonies. Out of the 355 colonies, only 5 colonies were identified as *Trichoderma*: 3 colonies from the paddy field (PDY1, PDY2, and PDY3); 1 colony from the rubber plantation (RBR4); and 1 colony from the banana plantation (BNN5).

Only two endophytic fungal colonies appeared from the weed leaves and were selected based on their morphological pigmentation and mycelium features; these were named Colony 1 and Colony 2.

### 3.2. Morphological and Molecular Identification

The seven total colonies, coded PDY1, PDY2, PDY3, RBR4, and BNN5 from soil (Figure 1A–C), and Colony 1 and Colony 2 from parthenium leaf (Figure 1D), were identified morphologically. For the soil colonies, we focused on *Trichoderma* to identify the species features, while for leaf colonies, we focused on random filamentous fungi. 

On the basis of the culture observation, the *Trichoderma* species had an irregular form, flat elevation, and an undulate margin, making them different from other fungal colonies. Other than that, *Trichoderma* colonies are greenish or yellowish green. The formation of the colonies was faster than that for other fungi, where they took less than five days to fully colonize the medium plates. The greenish or yellowish-green color usually appeared on day three or four of growth. On the other hand, the two colonies grown from leaves were pale whitish or blackish green with flappy-type mycelia.

The colors of colonies PDY1, PDY2, PDY3, RBR4, and BNN5, coded as UMK04, were all blue-green or yellowish-green to dark green. The elongation of mycelia had resulted in the full colonization of the medium plate at day four; white mycelia appeared in the center, which later changed completely from blue to dark green after a week of isolation. 

Conidiophores were erect and formed in variable branches. Phialides were lageniform in shape and more or less ampulliform (Figure 2). All these features showed the characteristics of *T. reesei*.

The features of leaf Colony 1 and leaf Colony 2, coded as UMK02, were a pale yellow-white color at the early stage, later turning into a dark brown or dark green color (Figure 3B). The conidiophores were long with phialides that contained conidia. Abundant circular conidia were observed under the light microscope (Figure 3A), which showed the colonies to be very similar to the genus *Aspergillus,* and a species feature match with *A. awamori*. 

The phylogenetic tree (Figure 4) clearly shows the affiliation of the colonies as *T. reesei* and *A. awamori*, in line with the morphological identification.

### 3.3. Analysis of Conidial Density in Trichoderma reesei and Aspergillus awamori

The viability of cells from conidia counting is important for fermentation, as well as the density of the population and the process of cell wall degradation. The concentration of viable cells for *T. reesei* was 3.8 × 10^6^ conidia/μL, while it was 1.908 × 10^6^ conidia/μL for *A. awamori*. From this study, we found that the number of viable cells for *T. reesei* was much higher than that for *A. awamori* (Figure 5).

### 3.4. Assay of Cellulase Enzyme Production 

The fungal cells of *T. reesei* and *A. awamori* were cultivated via submerged fermentation incorporated with rice straw to produce cellulase enzyme. The data on cellulase production were collected at the three time points of 3 days, 5 days, and 7 days. The cellulase enzyme production gradually increased from 3 days and 5 days in *T. reesei* cultivation, whereas in *A. awamori*, the cellulase production patterns were similar at 3 days and 5 days of cultivation, based on the glucose enzyme assay in different supernatant volumes of 100 μL, 200 μL, 250 μL, 300 μL, 400 μL, and 500 μL (Table 1). At 7 days of cultivation, in both fungi, production was decreased (Table 1). The cellulase production in rice straw incorporated with *T. reesei* showed that the concentration of enzyme at 3 days incubation time ranged from 5.56 U/mg to 18.89 U/mg, with the maximum at 500 μL of volume. The concentration of enzyme at 5 days incubation time also ranged from 12.22 U/mg to 27.04 U/mg, with the maximum at 500 μL of volume. The concentration of enzyme at 7 days incubation time ranged from 6.3 U/mg to 12.96 U/mg, with the maximum at 500 μL of volume. The results show that the best enzyme production by *T. reesei* incorporated with rice straw is 27.04 U/mg at 500 μL of volume for 5 days incubation time. The cellulase concentration in *A. awamori* at 3 days incubation time ranged from 2.96 U/mg to 15.56 U/mg, with the maximum at 500 μL of volume. The concentration of enzyme at 5 days incubation time also ranged from 6.3 U/mg to 15.19 U/mg, with the maximum at 500 μL of volume. The concentration of enzyme at 7 days incubation time ranged from 5.56 U/mg to 11.48 U/mg, with the maximum at 500 μL of volume. The results show that the best enzyme production by *A. awamori* incorporated with rice straw is 15.56 U/mg at 500 μL of volume for 3 days incubation time. A comparison between the fungi showed that cellulase production was significantly higher in *T. reesei* as compared to *A. awamori* in all supernatant volumes. It also showed that cellulase production varied, based on the supernatant volume, in both fungi (Figure 5).

## 4. Discussion

Different species have different activities. Thus, the isolation and identification of potential fungi are key to properly utilizing these species for human activities. In this study, we isolated fungi from soil and weed leaves. Their colonies showed similar morphological color, while in anatomical studies, they showed different shapes of conidia, different patterns of conidiophore branches, and different phialides. Therefore, on the basis of the anatomical studies, the two species were identified as *Trichoderma reesei* UMK04 (with the preliminary colony names of PDY1, PDY2, PDY3, RBR4, and BNN 5) from soil, and *Aspergillus awamori* UMK02 (preliminarily Colony 1 and Colony 2) from weed leaves. The morphological and anatomical characteristics of *T. reesei* UMK02 match with the findings of Asis et al. [19]. The morphological features of Colonies 1 and 2 were very similar to those of *A. awamori* UMK04, in accordance with a study by Oh et al. [20].

Fungi with a dense conidial population work faster in cellulose degradation, as observed with *T. reesei*, to produce cellulase enzyme. In this study, the number of viable cells for *T. reesei* was much higher than that for *A. awamori*; this is in line with the work of Al-Hazmi and Javeed [21], which showed that a higher density of *Trichoderma* sp. spores improved the growth of tomato. 

Cellulases are hydrolytic enzymes that are produced by microbes during the degradation process of cellulose or plant fibrous parts, and bacteria and fungi are good producers of cellulases [22]. In general, there are two types of fermentation techniques: solid-state fermentation (SSF) and submerged fermentation (SmF) processes. Both of these techniques have been widely used and studied in the production of cellulases [23]. In this study, two types of fungi—*T. reesei* from soil and *A. awamori*—were used for the determination of cellulase production using rice straw submerged culture. It is well-documented that cellulolytic microorganisms in the ascomycete group of *T. reesei* show the most potential for cellulase production [24]. The use of *Aspergillus* sp. in industrial processes is also well-documented. However, the *A. awamori* isolated in this study is the first to be isolated from Malaysian agriculture. Therefore, this newly isolated fungus of *A. awamori* was used for comparison with *T. reesei* regarding cellulase production. The results show that *T. reesei* produced a higher amount of cellulase enzyme than did *A. awamori* (Figure 5). The strong ability to respond to diverse environmental signals, and the fast growth of the fungus due to a high density of the conidial population, make *T. reesei* more effective, or faster, at degrading the cell wall of the available substrate [25]. 

## 5. Conclusions

The results of this study indicate that cellulase enzymes can be produced via the submerged fermentation of rice straw incorporated with *T. reesei* or *A. awamori* very quickly and without any nutrient supply. *T. reesei* is better for the production of cellulase enzyme (27.04 mg/0.5 mL) as compared with *A. awamori* (15.19 mg/0.5 mL). Among the three incubation times of 3, 5, and 7 days, 5 days of incubation was the best time for the production of a higher amount of cellulase enzyme using *T. reesei*, while for *A. awamori*, 3 days of incubation time was the best. In addition, using rice straw for cellulase production could reduce national rice straw waste, and it could benefit the enzyme industry in terms of the effective cost. 

## Figures and Tables

**Figure 1 jof-07-00868-f001:**
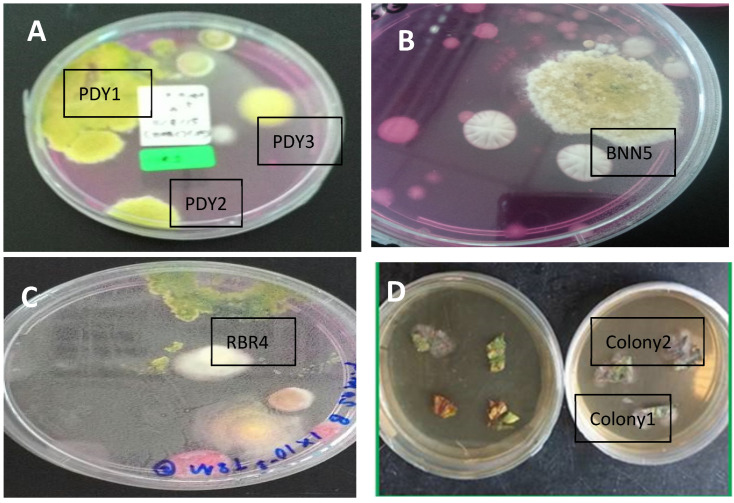
Isolation of fungal colonies. (**A**–**C**) Fungal isolation from soil cultures on DRBA medium; (**D**) Fungal isolation from Parthenium weed leaves on PDA medium.

**Figure 2 jof-07-00868-f002:**
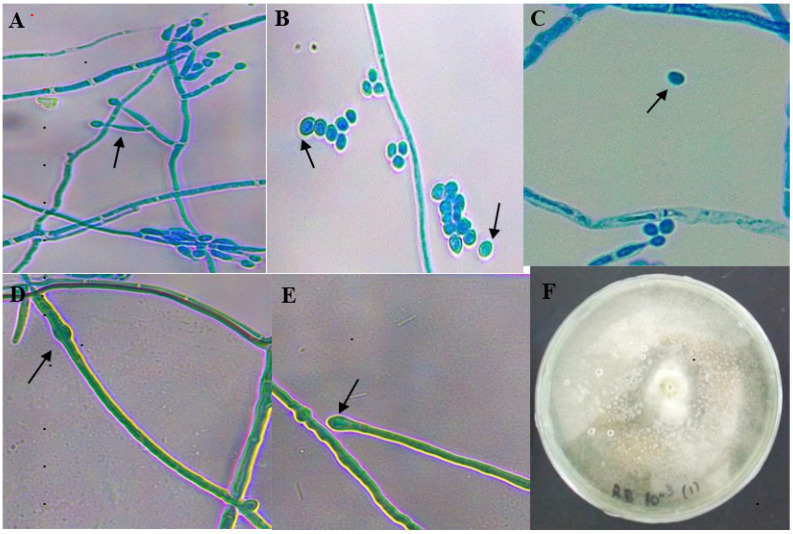
Morphological characteristic of *Trichoderma reesei*. a: unpaired conidiophores pattern (arrow), (**A**–**C**) subglobose to obovoid conidia (arrow), (**D**) chlamydosphores (arrow), (**E**) hyphal tip, (**F**) conidiation of *Trichoderma reesei* in plate. (Under microscope Leica: 40×).

**Figure 3 jof-07-00868-f003:**
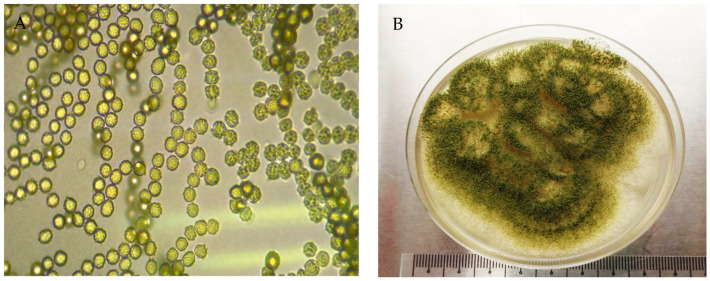
Morphological features under a light microscope (40×) of *Aspergillus awamori* isolated from Parthenium weed leaves. (**A**): Phialospores arranged in two rows, (**B**): Full colony growth on a PDA plate.

**Figure 4 jof-07-00868-f004:**
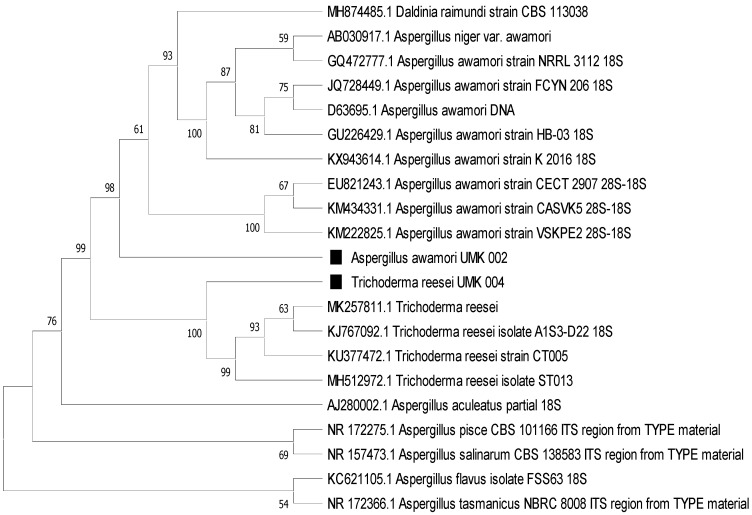
Phylogenetic identification of the colonies of UMK04 and UMK02 using maximum likelihood in MEGA-X. The phylogenetic tree was generated in MEGA-X [11] using maximum parsimony analysis with 1000 bootstrap replications. The branches indicate bootstrap coefficients of >50%.

**Figure 5 jof-07-00868-f005:**
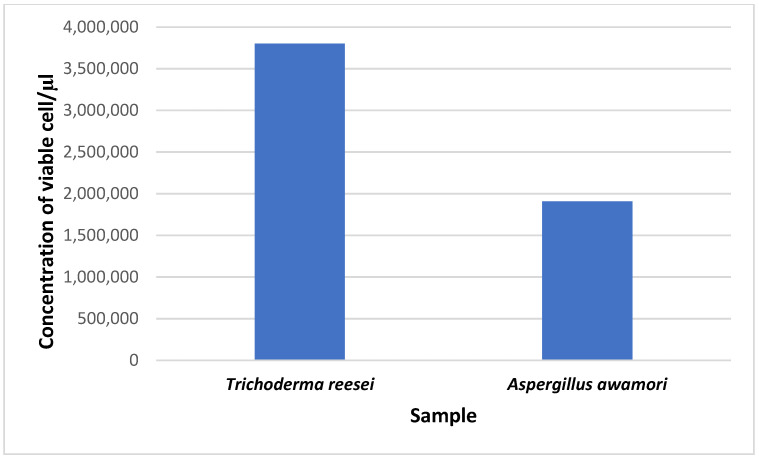
The conidia count in seven-day-old cultures of *T. reesei* and *A. awamori*.

**Table 1 jof-07-00868-t001:** Mean difference multiple comparison tests of the cellulase enzyme concentration in *T. reesei* and *A. awamori* fermented with rice straw for incubation periods of 3, 5, and 7 days. Asterisks (*) indicate significant differences at the 0.05 level.

Days	Volume (μL)	Concentration (U/mg)		Significant
		*T. reesei*	*A. awamori*	
3 days	100	5.56 ± 3.65	2.96 ± 1.75	0.000 *
	200	11.48 ± 5.18	3.70 ± 2.67	0.000 *
	250	11.85 ± 5.14	10.00 ± 1.85	0.000 *
	300	16.67 ± 6.92	11.85 ± 2.46	0.000 *
	400	17.04 ± 7.15	14.81 ± 2.72	0.000 *
	500	18.89 ± 7.06	15.56 ± 2.25	0.000 *
5 days	100	12.22 ± 3.65	6.30 ± 1.75	0.000 *
	200	17.04 ± 5.18	8.89 ± 2.67	0.018 *
	250	20.37± 5.14	11.85 ± 1.85	0.000 *
	300	25.19 ± 6.92	13.33 ± 2.46	0.000 *
	400	26.30 ± 7.15	13.70 ± 2.72	0.000 *
	500	27.04 ± 7.06	15.19 ± 2.25	0.000 *
7 days	100	6.30 ± 3.65	5.56 ± 1.75	0.000 *
	200	6.67 ± 5.18	7.41 ± 2.67	0.000 *
	250	11.11 ± 5.14	8.15 ± 1.85	0.000 *
	300	11.48 ± 6.92	8.52 ± 2.46	0.000 *
	400	12.22 ± 7.15	9.63 ± 2.72	0.000 *
	500	12.96 ± 7.06	11.48 ± 2.25	0.000 *

## Data Availability

Not applicable.

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
