# Peer review of "Cellulase Enzyme Production from Filamentous Fungi *Trichoderma reesei* and *Aspergillus awamori* in Submerged Fermentation with Rice Straw"

_jof, 2021, doi:10.3390/jof7100868_

Round 1
Reviewer 1 Report
The authors need to revise the introduction ( previous work, shortcomings, solution and the new thing proposed by the authors). At the present state, it is not appropriate for a scientific paper. The conclusion is insufficient and does not showcase any new findings. What does this '*' represent in Table 1?
Improve the presentation of the paper (Equation A) (line 254). Provide validation of results and standard deviation of experimental values.
Why have you carried out this study and what would be the impact of your findings on societal development?
Author Response
Point 1: The authors need to revise the introduction ( previous work, shortcomings, solution and the new thing proposed by the authors). At the present state, it is not appropriate for a scientific paper. The conclusion is insufficient and does not showcase any new findings.
Response 1: We have edited the paper focus on problem statement and novelty.
What does this '*' represent in Table 1?
Response 1: Edited.
Improve the presentation of the paper (Equation A) (line 254). Provide validation of results and standard deviation of experimental values.
Response 1: Edited the equation. Sorry not clear about the comments, we already have included the standard deviation.
Why have you carried out this study and what would be the impact of your findings on societal development?
Response 1: Cellulase production is costly in especially for the degradation of cell wall, so researcher is finding to easy way to the degradation process and microbes is one of the elements to degrade the plant cell wall faster that chemical process. In industrial society will be benefited if can identify more promising microbes. In the paper we have focused these issues.
Reviewer 2 Report
Through out paper there are corrections, References not as per journal format. Many of the section without references. The paper is a preliminary studies which is not suitable for the "Journal of Fungi"
Author Response
Through out paper there are corrections, References not as per journal format. Many of the section without references. The paper is a preliminary study which is not suitable for the "Journal of Fungi"
Response: Thank you for the comments. We have edited the paper for the improvements of language, grammar and refences. In this paper reported the first isolation of Aspergillus awamori in Malaysian agriculture, which evaluated by morphological and molecular studies as well as also tested as cellulase production as for potential application, so we would like to inform that it’s not really a preliminary study. We would be grateful if you consider our paper.

Reviewer 3 Report
Dear authors,
The manuscript is a simple one that has the potential to be published in “Journal of Fungi”, but it is mandatory to make improvements. Below are the general comments and specific comments, the point-by-point recommendations to be implemented.
General comments:
The massive generation of waste or by-products has made their disposal an important challenge to overcome. The manufacturing of enzymes is expensive, leading to high costs of commercially available enzyme mixtures, raising the production costs of hydrolysis processes. In recent years, the interest in cellulase has increased due to the numerous potential applications for these enzymes. Considering these things, the topic selected by the authors has potential.
However, the paper has many sentences that are difficult to read. A professional English language editing service is strongly recommended. Also, many mistakes are found throughout the manuscript that needs to be corrected (text manuscript, sentences, references, etc.). Moreover, some methodological details will improve the manuscript further. In the end, the conclusion needs to be refined by the authors. Furthermore, the most important, what new things have you done in this work since many papers are already available on a similar topic?
Specific comments:
- Title: “Cellulase enzyme production from filamentous fungi Trichoderma reseei and Aspergillus awamori incorporated with rice straw”. Incorporated? I recommend changing the title.
- I recommend checking the author list, especially for “M A Halim Sheikh” and “M Rezaul Karim”.
- Line 28 – “…of 3 days, 5 days, and 7 days” Change to “of 3, 5, and 7 days”.
- Line 35 – “ reesei mycelia degraded more cell wall of rice straw…”. T. reesei mycelia degraded? I recommend modifying the sentence.
- Lines 44, 57, etc. – Change “byproduct” to “by-product” Also, check the manuscript and correct the word.
- Line 44 – “Discovery of new fungal species…” – What is new reesei or A. awamori? I recommend modifying the sentence.
- Line 49 – “…fungal cellulases to digest lignin, hemicelluloses, and cellulose” Cellulases are a group of enzymes that catalyze the degradation of cellulose, a polysaccharide built of β-1,4 linked glucose units (https://doi.org/10.1016/0038-0717(94)90216-X) not lignin or other matrices. I recommend correcting the sentence.
- Line 57 – Change the sentence to “Rice straw is a lignocellulose component consisting of cellulose, hemicellulose, and lignin.” Also, I recommend mentioning the percentages for the three compounds (cellulose, hemicellulose, and lignin) in the rice straw matrix.
- Line 59 – “Cellulose is important compound that produce cellulase enzyme.” Cellulases are not produced by cellulose. Fungi or other microorganisms produce cellulase but in no case cellulose. Cellulose is a matrix. Please correct this mistake.
- Lines 47 to 48 and 59 to 61 – This two-section repeats the same things “various industries such as laundry and detergent, textile production, pulp, and paper, in the food industry…” and “…various sectors including beverage, food, detergent, pulp, paper, textile…”. I recommend that you avoid repeating the same content. This problem was also observed throughout the manuscript. Please modify.
- Line 64 – Change the “microbe” word to “microorganisms”.
- The introduction part must be modified. In addition to the problems identified, it should also be mentioned that it is very short and does not mention the element of novelty desired with this article.
- Line 77 – “Trichoderma” – italic format. All latin words/names will be written in italic format. This has not been verified in the manuscript. I recommend that you check the manuscript and keep this in mind.
- Line 77 – Change “grams” to “g”.
- Line 70 – Is it a new paragraph?. I recommend checking.
- Line 110 – Add a space between sentences “…culture [7].The…”. “Trichoderma” – Italic format.
- Lines 108 and 109 – “10−1, 10−2, 10−3, 10−4, and 10−5”. Put the comma usually, not as a Superscript.
- Line 117 – “”. Add a space.
- I recommend that you modify subchapter 2.6. It also has another subchapter, and it is ambiguous. Also, there is an empty line (144), which I recommend you change.
- Line 148 – I recommend mentioning what microscope was used (name/model, country, city).
- Line 149 – “Aneens” change to Aneens et al.
- Line 161 – “mix,,” Delete a comma.
- Line 163 – “mins” change to “min”.
- Line 219 – “10^6” change to “106”.
- Line 224 – “The broth culture containing 30 g of rice straw…” – I recommend mentioning if the authors made a mix between broth and rice straw or just rice straw. It is not clear.
- Chapter 2.11 – What protocol was used? I recommend mentioning a reference for this protocol. In addition, I recommend consulting this new article (https://doi.org/10.3390/jof7090766) for more information to complete in this subchapter (2.12) and also for 12 Enzyme assay analysis.
- Line 251 – Ref. 13 is H., Rosebrough, N. J., Farr, A. L., & Randall, R. J. (1951). Protein measurement with the Folin phenol reagent. Journal 439 of biological chemistry, 193, 265-275, and in the text is enzyme activity, and there is no connection between topics.
- Chapter 2.10 and chapter 2.11.
- I recommend considering this review article “Teigiserova, D.A.; Bourgine, J.; Thomsen, M. Closing the loop of cereal waste and residues with sustainable technologies: An overview of enzyme production via fungal solid-state fermentation. Sustainable Production and Consumption 2021, 27, 845-857, doi:10.1016/j.spc.2021.02.010.” and “The unit U/gds represents “enzyme unit per gram of initial dry solid substrate.” The main advantage of this unit is that it connects the product’s efficiency to the quantity of substrate used, giving an evaluation of the material recovery potential. This unit is the most comparable and applicable across disciplines due to the clear connection to the quantification of the feedstock used (dry matter enables comparison to other types of waste as opposed to the fresh biomass), and already established usage appearing in more than 50% of the articles.” I strongly recommend calculating the enzymatic activity as a unit / g of dry substrate used or as mentioned above, not mg/ml.
- In addition, I strongly recommend that authors use this article (Martău, G.-A., Unger, P., Schneider, R., Venus, J., Vodnar, D.C., López-Gómez, J.P. 2021. Integration of Solid State and Submerged Fermentations for the Valorization of Organic Municipal Solid Waste. Journal of Fungi, 7(9), 766) for cellulase activity assay.
- Line 233 – “sodium citrate buffer” – I recommend mentioning the concentration and pH of the buffer.
- I recommend adding equation A as a figure or as a text. Now in the manuscript is a picture with a text (different font and text style).
- It is very important to mention whether the experiments were performed in triplicate or not, especially for the enzymatic activity (this is missing).
- Figure 4. The concentration of viable cells – I recommend using a logarithmic value. Also, mane of bacteria, using an italic text.
- Line 335 – “whilst” ? check it.
- Table 1 – The “Significant” column must be merged into the value column, and the asterisk (*) must explain what it means in the table.
- Lines 392 – 293 – This thing has already been said in the introduction.
Finally, the article has potential, but to be published in the Journal of Fungi, it is mandatory to check the entire manuscript in terms of the English language and its organization according to the journal template/guideline.
Author Response
Point 1. Title: “Cellulase enzyme production from filamentous fungi Trichoderma reseei and Aspergillus awamori incorporated with rice straw”. Incorporated? I recommend changing the title.
Response 1: Cellulase enzyme production from filamentous fungi Trichoderma reseei and Aspergillus awamori in submerged fermentation with rice straw
Point 2: I recommend checking the author list, especially for “M A Halim Sheikh” and “M Rezaul Karim”.
Response 2: Sorry, not clear about it, but tried to corrected.
Point 3 : Line 28 – “…of 3 days, 5 days, and 7 days” Change to “of 3, 5, and 7 days”.
Response 3: Corrected.
Point 4: Line 35 – “ reesei mycelia degraded more cell wall of rice straw…”. T. reesei mycelia degraded? I recommend modifying the sentence.
Response 4: Corrected.
Point 5: Lines 44, 57, etc. – Change “byproduct” to “by-product” Also, check the manuscript and correct the word.
Response 5: Corrected.
Point 6: Line 44 – “Discovery of new fungal species…” – What is new reesei or A. awamori? I recommend modifying the sentence.
Response 6: Actually, it should be researcher…which has corrected.
Point 7: Line 49 – “…fungal cellulases to digest lignin, hemicelluloses, and cellulose” Cellulases are a group of enzymes that catalyze the degradation of cellulose, a polysaccharide built of β-1,4 linked glucose units (https://doi.org/10.1016/0038-0717(94)90216-X) not lignin or other matrices. I recommend correcting the sentence.
Response 7: Corrected.
Point 8: Line 57 – Change the sentence to “Rice straw is a lignocellulose component consisting of cellulose, hemicellulose, and lignin.” Also, I recommend mentioning the percentages for the three compounds (cellulose, hemicellulose, and lignin) in the rice straw matrix.
Response 8: Corrected.
Point 9: Line 59 – “Cellulose is important compound that produce cellulase enzyme.” Cellulases are not produced by cellulose. Fungi or other microorganisms produce cellulase but in no case cellulose. Cellulose is a matrix. Please correct this mistake.
Response 9: Corrected.
Point 10: Lines 47 to 48 and 59 to 61 – This two-section repeats the same things “various industries such as laundry and detergent, textile production, pulp, and paper, in the food industry…” and “…various sectors including beverage, food, detergent, pulp, paper, textile…”. I recommend that you avoid repeating the same content. This problem was also observed throughout the manuscript. Please modify.
Response 10: In this section mentioned ‘As mentioned earlier” and also deleted the repetition terms.
Point 11: Line 64 – Change the “microbe” word to “microorganisms”.
Response 12 : Corrected.
Point 13: The introduction part must be modified. In addition to the problems identified, it should also be mentioned that it is very short and does not mention the element of novelty desired with this article.
Response 13 : Added the it.
Point 14: Line 77 – “Trichoderma” – italic format. All latin words/names will be written in italic format. This has not been verified in the manuscript. I recommend that you check the manuscript and keep this in mind.
Response 14 : Corrected.
Point 15: Line 77 – Change “grams” to “g”.
Response 15 : Corrected.
Point 16: Line 70 – Is it a new paragraph?. I recommend checking.
Response 16 : Corrected.
Point 17: Line 110 – Add a space between sentences “…culture [7].The…”. “Trichoderma” – Italic format.
Response 17 : Corrected.
Point 18: Lines 108 and 109 – “10−1, 10−2, 10−3, 10−4, and 10−5”. Put the comma usually, not as a Superscript.
Response 18: Corrected.
Point 19: Line 117 – “”. Add a space.
Response 19: Sorry not found.
Point 20: I recommend that you modify subchapter 2.6. It also has another subchapter, and it is ambiguous. Also, there is an empty line (144), which I recommend you change.
Response 20: Corrected.
Point 21: Line 148 – I recommend mentioning what microscope was used (name/model, country, city).
Response 21: Corrected.
Point 22: Line 149 – “Aneens” change to Aneens et al.
Response 22: Corrected.
Point 23: Line 161 – “mix,,” Delete a comma.
Response 23: Corrected.
Point 24: Line 163 – “mins” change to “min”.
Response 24: Corrected.
Point 25: Line 219 – “10^6” change to “106”.
Response 25: Corrected.
Point 26: Line 224 – “The broth culture containing 30 g of rice straw…” – I recommend mentioning if the authors made a mix between broth and rice straw or just rice straw. It is not clear.
Response 26: Corrected.
Point 27: Chapter 2.11 – What protocol was used? I recommend mentioning a reference for this protocol. In addition, I recommend consulting this new article (https://doi.org/10.3390/jof7090766) for more information to complete in this subchapter (2.12) and also for 12 Enzyme assay analysis.
Response 27: Corrected. We already used the methods from Zhang et al 2009 (https:// doi: 10.1007/978-1-60761-214-8_14)
Point 28:Line 251 – Ref. 13 is H., Rosebrough, N. J., Farr, A. L., & Randall, R. J. (1951). Protein measurement with the Folin phenol reagent. Journal 439 of biological chemistry, 193, 265-275, and in the text is enzyme activity, and there is no connection between topics.
Response 28: Sorry for wrong citation. Corrected.
Point 29: Chapter 2.10 and chapter 2.11.
I recommend considering this review article “Teigiserova, D.A.; Bourgine, J.; Thomsen, M. Closing the loop of cereal waste and residues with sustainable technologies: An overview of enzyme production via fungal solid-state fermentation. Sustainable Production and Consumption 2021, 27, 845-857, doi:10.1016/j.spc.2021.02.010.” and “The unit U/gds represents “enzyme unit per gram of initial dry solid substrate.” The main advantage of this unit is that it connects the product’s efficiency to the quantity of substrate used, giving an evaluation of the material recovery potential. This unit is the most comparable and applicable across disciplines due to the clear connection to the quantification of the feedstock used (dry matter enables comparison to other types of waste as opposed to the fresh biomass), and already established usage appearing in more than 50% of the articles.” I strongly recommend calculating the enzymatic activity as a unit / g of dry substrate used or as mentioned above, not mg/ml.
In addition, I strongly recommend that authors use this article (Martău, G.-A., Unger, P., Schneider, R., Venus, J., Vodnar, D.C., López-Gómez, J.P. 2021. Integration of Solid State and Submerged Fermentations for the Valorization of Organic Municipal Solid Waste. Journal of Fungi, 7(9), 766) for cellulase activity assay.
Response 29: Our calculation based on regression line standard curve equation. We have corrected as U/mg. The above-mentioned calculation, sorry we are not clear about the method.
Point 30: Line 233 – “sodium citrate buffer” – I recommend mentioning the concentration and pH of the buffer.
Response 30: Corrected
Point 28: I recommend adding equation A as a figure or as a text. Now in the manuscript is a picture with a text (different font and text style).
Response 30: Corrected
Point 31: It is very important to mention whether the experiments were performed in triplicate or not, especially for the enzymatic activity (this is missing).
Response 31: Corrected
Point 32: Figure 4. The concentration of viable cells – I recommend using a logarithmic value. Also, mane of bacteria, using an italic text.
Response 32: Here we just report the only at seven days culture spore density between the two fungi. We not carried out throughout seven days of growth, therefore, cannot show the logarithm value.
Point 33: Line 335 – “whilst” ? check it.
Response 33: Corrected as whereas
Point 34: Table 1 – The “Significant” column must be merged into the value column, and the asterisk (*) must explain what it means in the table.
Response 34: Corrected.
Point 35: Lines 392 – 293 – This thing has already been said in the introduction.
Response 35: Corrected.
Finally, the article has potential, but to be published in the Journal of Fungi, it is mandatory to check the entire manuscript in terms of the English language and its organization according to the journal template/guideline.
Response: Thank for critically reviewing the paper. As for improvement of English language, we have edited with English expert and also improved the format. We would be grateful if you favorable consider the paper to publish in Journal of Fungi.
Round 2
Reviewer 1 Report
It could be considered for publication.
Author Response
Dear Reviewer,
Thank you for your consideration. We really appreciate it.
Regards
Reviewer 2 Report
No Comments
Author Response
Dear Reviewer,
Thank you for reviewing the paper. We will be grateful, if you consider our paper to publish in MDPI.
Reviewer 3 Report
The authors tried to respond promptly to all comments, but the manuscript still has many errors identified. An example and the most relevant is even in the title; authors in "response to reviewer" have a different title compared to the title in the manuscript. This thing leads me to think that the authors reviewed the article very superficially. In addition, the authors still did not follow the template and the journal guidelines for authors (examples are the chapters and subchapters format/style, references, etc.).
In addition, the unit U/gds represents “enzyme unit per gram of initial dry solid substrate”. In this case, the value from Table 1 is the same as the previous version of the manuscript. Totally wrong! The U/gds unit represents "enzyme unit per gram of initial dry solid substrate". This value must be calculated from the results obtained.
I recommend that you use this protocol https://www.nrel.gov/docs/gen/fy08/42628.pdf to express enzymatic activity for cellulase.
More comments can be found in the attached document. The manuscript cannot be published in this form until these problems are resolved.

Author Response
Dear reviewer,
Thanks for critically review the paper.
Point 1: The most relevant is even in the title; authors in "response to reviewer" have a different title compared to the title in the manuscript.
Response 1: Thank you for the comment and sorry for the inconvenience. Already corrected the title as Cellulase enzyme production from filamentous fungi Trichoderma reesei and Aspergillus awamori in submerged fermentation with rice straw’ in the manuscript.
Point 2: The authors still did not follow the template and the journal guidelines for authors (examples are the chapters and subchapters format/style, references, etc.).
Response 2: So far looks we have corrected and followed the format but currently, JoF now accepts free format submission. We will appreciate if JoF indicates the exact wrong format.
Point 3: The unit U/gds represents “enzyme unit per gram of initial dry solid substrate”. In this case, the value from Table 1 is the same as the previous version of the manuscript. Totally wrong! The U/gds unit represents "enzyme unit per gram of initial dry solid substrate". This value must be calculated from the results obtained.
Response 3: Actually, the previous one mistakenly put wrong unit for expression (mg/ml), which corrected in the revised version (U/mg of dry weight straw). As previous version and current version both time We stated that used regression line equation to calculate the amount, as followed the method in Nahar et al., 2012. Therefore, calculation is same as previous. If you want to see our raw data, we can send it.
Point 4: Commented not clear ‘The enzyme production was also carried out at different incubation times of 3, 5, and 7 days of culture with the fermented rice straw’.
Response 3: Actually, the language was checked by MDPI editing service. However, we have tried again to clear.
Point 5: mg of what? Dry weight?
Response 5: Corrected.
Point 6: Cellulose is a substrate and do not produce cellulase.
Response 6: Corrected.
Point 7: It is not clear what the authors want to say.
Response 7: Corrected.
Point 8: Incomplete sentence.
Response 8: Corrected
Point 9: Rice straw or cellulose extraction from rice straw?
Response 10: Corrected.
Point 11: mg of what? Substrate? or what?
Response 12: Corrected.
Point 12: Figure 5.
Respnse 12: Sorry not understood, do we need to make italic?
Point 13: Figure 4.
Response 13: Corrected.
